# The hidden/ hard to reach men who have sex with men (MSM)—Results from the qualitative study in India

Seema Sahay[1,2*], Dhammasagar Ujagare[1,3], Girish Rahane[1], Tuman Lal Katendra[1], Amarendra Mahapatra[4], Shivendra Kumar Singh[5], Sanjeev Kumar[6], Chinmoyee Das[7], Bhawani Singh Kushwaha[7], Vinita Verma[7], Rajiv Ranjan Tiwari[7], Nupur Mahajan[7], P.S. Saravanamurthy[8], Bitra George[8], Sampada Bangar[1,3]

1 ICMR- National Institute of Translational Virology & AIDS Research (ICMR-NITVAR) [Formerly, ICMR-NARI], Pune, India, 2 Academy of Scientific and Innovative Research (AcSIR), Ghaziabad, India, 3 Savitribai Phule Pune University, Pune, India, 4 ICMR- Regional Medical Research Center, Bhubaneswar, India, 5 Upgraded Department of Community Medicine and Public Health, King George's Medical University, Lucknow, India, 6 Department of Community & Family Medicine, All India Institute of Medical Sciences, Bhopal, India, 7 National AIDS Control Organization, New Delhi, India, 8 FHI360 Country office, Linkages-India, New Delhi, India

* seemasahay99@gmail.com

## Abstract

### Background

To ensure a substantial impact and prevent the resurgence of the epidemic, the HIV programmes should effectively reach hidden key populations (KPs). A qualitative study was conducted to understand hidden men who have sex with men (MSM), and their needs and identify outreach strategies for those who were still unreached.

### Methods

Between 17 March 2019 and 26 July 2019, 42 In-depth Interviews (IDIs), 40 Key Informant Interviews (KIIs), and 16 Focus Group Discussions (FGDs) were conducted covering 170 participants across four states of India. Data were audio recorded, transcribed, translated and analyzed using NVivo software employing a grounded theory approach.

### Results

A diverse spectrum of unreached men were reportedly practicing same-sex behaviors and new hidden MSM categories have been identified. These hidden MSM demonstrated denial, lack of awareness, and a history of forced sexual experiences. Being elusive, hence no access to healthcare services and education resulted in condom-less sex. The very young adolescent MSM were unaware of the risks and perceived risky practices as the norm. Both at urban and rural settings, MSM

**Data availability statement:** Data cannot be shared publicly because data contains potentially identifying and sensitive information of the study participants. Data is available at centralized data repository system at the ICMR-NITVAR. Data will be available on request to any researcher. The contact details are; director@nariindia.org and seemasahay99@gmail.com.

**Funding:** This work was supported by the United States Agency for International Development (USAID) through FHI 360/Linkages [Grant Number: AID-OAA-A-14-00045]. The funders had no role in study design, data collection and analysis, decision to publish, or preparation of the manuscript.

**Competing interests:** The authors have declared that no competing interests exist.

respondents shared about sexual abuse by a male while studying in 2nd standard to 6th standard. The 'sugar daddies/ older MSM' who were >50 years old get disengage from the programme, preferentially practice high-risk behaviors, abusing young boys and men who fall prey for money or easy fun. Peers were the most acceptable persons to reach 'hidden' MSM.

## Conclusion

A peer-driven approach based on the inbuilt trust of the community and leveraging networks might be a vital outreach strategy to engage this elusive, vulnerable hidden MSM. The vulnerability of young boys and men calls for an urgent need for fostering safer environment for minor male children in the context of this study and strong integration of POCSO Act in the programme. Additionally, there is an urgent need to revisit strategies for engaging middle-aged MSM, the 'sugar daddies' or 'uncles' in preventing spread of HIV. Policymakers and educational institutions may prioritize comprehensive sex and sexuality education to challenge and transform prevailing social norms affecting young boys in India. It is important for parents and carers to be alert about how these abuse happen, and how to help prevent it.

## Introduction

Globally, men who have sex with men (MSM) experience a significantly higher risk of HIV infection. The World Health Organization (WHO) reports that the likelihood of contracting HIV is 26 times greater among MSM compared to the general population [1]. In 2022, MSM made up 67% (21,400) of the estimated 31,800 new HIV infections, and 87% of the estimated infections among all males in United States. Compared to 2018, the annual number of HIV infections among MSM decreased by 10% in 2022 [2]. Despite overall declines in HIV incidence in some regions, MSM continue to experience high rates of new infections.

With estimated 2.3 million people living with HIV in 2021 in India and 68 thousand infected newly [3], India becomes the third highest HIV burden country in the world. Globally, including India, MSM, Transgender Women (TGW), and non-client partners of key populations (KPs) strongly contributed to the proportions of adult new HIV infections in 2022 compared with 2010 [4]. It has been anticipated that with growing access to HIV services and overall decreasing HIV incidence, HIV would concentrate more in core group of KPs and relative risk of HIV infection among KPs may increase [5,6]. The national prevalence of HIV among the general population is 0.20% while the recent HIV prevalence among MSM and Hijra/ Transgender (H/ TG) persons has been documented to be 3.26% and 3.78% respectively [7]. In context of SDG goals of reducing HIV infection by 90%, attaining such reductions in low-level epidemic settings such as India, has been an envisaged challenge [8]. To achieve a significant impact and prevent the reversal of epidemic, it is essential that HIV programme reach these hidden KPs. In the absence of PrEP or high rates of viral load suppression,

condom use among gay men and other MSM must be 70–80% to stabilize the epidemic in this population [9], a level reached in only a few countries.

Reaching MSM who are hidden/ hard to reach, remains challenging due to multiple vulnerabilities including extensive stigma [10]. In India, reaching adolescent MSM and Panthis (sexually penetrative MSM) has remained a challenge [11]. Hidden or hard to reach populations are socially invisible, floating, and non-homogenous who conceal their identity owing to socio-legal issues. The issue of trust and biases keep emerging against various strategies used to reach hidden MSM. This paper is a synthesis of evidence generated on prevalent psychosocial issues among hidden MSM, risks experienced, their networking, drivers of HIV risk, and mental health issues.

## Materials and methods

A multi-centric qualitative exploratory study was conducted in four states of India viz. Maharashtra (MH) [Pune (urban); Karad and Ichalkaranji (rural)] in Western India, Uttar Pradesh (UP) [Lucknow (urban); Barabanki (rural)] in Northern India, Madhya Pradesh (MP) [Bhopal (urban); Hoshangabad (rural)] in Central India, and Odisha (OR) [Bhubaneswar (urban); Sundargarh, Angul, and Banki (rural)] in Eastern India. MSM population categories recruited, and recruitment strategies are described in our previous paper [11]. The topic guides, study tools, and participant recruitment plan were co-developed in local vernacular languages (Hindi, Marathi, and Odiya) with the site teams and local Community Advisory Boards (CABs). The study was initiated in November 2018 and completed in September 2019 while between 17 March 2019 and 26 July 2019 data collection was completed. Following purposive and convenience sampling techniques, 42 In-depth Interviews (IDIs), 40 Key Informant Interviews (KIIs), and 16 Focus group discussions (FGDs) (n = 88) leading to a total coverage among 170 participants, were conducted. The IDIs were conducted with adult hard-to-reach MSM (MSM who were not reached by Targeted Interventions (TIs) ever) or hidden MSM. The KIIs were conducted with the primary and secondary stakeholders which included researchers, clinicians, NGO representatives, leaders of the support groups, and programme personnel. Since FGD was a group discussion, MSM who were visible and volunteered themselves, they were invited to participate. FGD participants were counselled not to talk about their own personal issues to avoid any kind of disclosure. The facilitator was trained to divert discussions to prevent disclosure of personal information. Risk of breach of confidentiality was discussed and explained. The FGD participants were *Kothis* (sexually receptive MSM), group of *Panthis* (sexually penetrative MSM) along with versatile and bisexual men. The study tools focused on social and sexual networks, community characteristics, concerns of hidden MSM, and barriers to accessing HIV prevention or health services. The rate of drop outs between scheduling interview and actually conducting interview was 5:2 (40%). The audio data were collected at the sites through face-to-face interviews or discussions by trained master-level social science researchers. IDI/KII/FGD lasted for 45–90 minutes. Each participant was given a unique identification number which included alphabet/s abbreviations to identify state, i.e., MH (Maharashtra), MP (Madhya Pradesh), UP (Uttar Pradesh) and OR (Odisha), U (Urban) or R (Rural) setting and type of data collection tool used (IDI/ KII/ FGD), numeric interview number, respondent category and age in years.

### Data analysis

The audio data were transcribed verbatim, translated into English, and typed in Microsoft Word. The translated electronic data from the sites were reviewed by two social scientists and the principal investigator in real-time for missing segments or identifying need for repeat interviews. Repeat interviews were conducted in case of missing/ additional information at sites. Data were analysed using N-Vivo release 1.7.2 (1560) (1999–2022) software. A data analysis workshop was conducted with site investigators to identify initial codes using representative segments and obtain reliable contextual findings. After further iterative readings emerging codes were combined into different domains which explained the layered nuances emerging from the data with relations to other categories. The final themes were identified based on the categories formed following further repeated iterations using the grounded theory approach [12]. The relationship between the themes helped in developing the analytical framework.

**Ethical considerations**

The study was approved by the Institutional Ethics Committee of the Implementing Research Institute, NACO's Technical Resource Group (TRG) for Research, NACO's Ethics Committee, Protection of Human Subjects Ethics Committee (PHSC) of FHI 360, and the Ethics Committees of all the participating research institutes. Written informed consent was obtained from all study participants for participation and audio recording.

## Results

The socio demographic profile (Table 1) shows that most of the participants self-identified themselves as Kothi (43%−51%) while one-fifth were reportedly double decker. Half of the participants resided in urban areas and three-fourths reported Hindu as their religion. All participants were literate while 60% of the IDI participants were graduate and above. More than 70% of the participants were unmarried, and 60% IDI and two-third FGD participants reported living in joint family. Almost 20% of the participants were students.

Both key MSM and NGO representatives accounted for 50% of the Key informants, followed by programme personnel (23%), counsellors (15%) and doctors (5%). Equal representation of urban and rural residence was maintained. More than three quarters reported Hindu as their religion and having graduation and above education. [Table 2].

The emerging data showed that the reasons for the disproportionate HIV burden among MSM are heterogeneous which includes biological factors, internalised and enacted stigma leading to behavioural challenges; and substance use to cope with a non-affirming society. The analytical framework shows the hidden MSM as hub with spokes showing newly identified demographics of hidden/ new MSM (Fig 1). All types of hidden MSM demonstrated various personality, risks and vulnerabilities viz. carefree attitude, denial, ignorance, seeking pleasure, fear, mental health issues, substance abuse and violence. The risk behaviours such as condom-less sex, sex with multiple partners and sex under intoxication encompass the above mentioned risks and vulnerabilities.

A total of 310 codes were generated initially, and final emerging themes are described. The final emerging themes were (1) Limited group networking facilitates secrecy of the group (2) Condom-less sex in hidden MSM (3) Expanding demographics of hidden MSM (4) Reaching the hidden MSM (5) Needs. Our study shows a number of newer demographics of men who might be 'MSM' although denying to be an MSM; hence the 'hidden MSM'.

**Limited group networking facilitates secrecy of the group**

The stigmas both enacted and self, the fear of discrimination, fear of being left alone and ensuing violence necessitates closed networking. Emerging theme 'limited group networking facilitates secrecy of the group' deals with the mechanism to have covert contacts. The MSM thrive in their own concentrated groups forming the network where preferential mixing occurs.

*"It is like they have their own group; they only prefer to meet those people who are like them..."* [##MP-U-KII-03##, programme personnel]

While travelling on the same routes, same people meet, network and they remain hidden.

*"[/I am/] in the marketing field, I travel, out of (=Pune=) to (=Sangli, Satara, Goa, Karnataka=)… then I get to meet [/ partners/]."* [##MH-U-IDI-03##, panthi, 40yrs]

Small, concentrated groups with preferential mixing of kins exist indicating circulation in small groups such as *'=name of city= group'* or *'Khandagiri sex group'*.

**Table 1. Socio-demographic information of IDI and FGD participants.**

| Characteristics | Response | IDI N = 42 (%) | FGD N = 88 (%) |
|---|---|---|---|
| **Age** | Mean (SD) | 26 (6.57) | 31 (10.17) |
| **Category (self-identified)** | Kothi | 18 (43) | 45 (51) |
| | Panthi | 6 (14) | 10 (11) |
| | Double decker | 9 (21) | 19 (22) |
| | Bisexual | 8 (19) | 11 (13) |
| | Cis-gender gay | 1 (2) | – |
| | Top | – | 1 (1) |
| | Refused to answer | – | 2 (2) |
| **Study state** | Maharashtra (MH) | 11 (26) | 22 (25) |
| | Madhya Pradesh (MP) | 10 (24) | 21 (24) |
| | Odisha (OR) | 10 (24) | 20 (23) |
| | Uttar Pradesh (UP) | 11 (26) | 25 (28) |
| **Residence** | Urban | 22 (52) | 47 (53) |
| | Rural | 20 (48) | 41 (47) |
| **Religion** | Hindu | 32 (76) | 64 (73) |
| | Muslim | 4 (10) | 17 (19) |
| | Other (Buddhist, Jain) | 1 (2) | 3 (3) |
| | Refused to answer | 5 (12) | 4 (5) |
| **Highest education completed** | Primary | 1 (2) | 8 (9) |
| | High school | 6 (14) | 26 (30) |
| | Higher secondary | 10 (24) | 28 (32) |
| | Diploma | – | 1 (1) |
| | Graduate | 18 (43) | 14 (16) |
| | Post graduate and above | 7 (17) | 9 (10) |
| | Refused to answer | – | 2 (2) |
| **Marital status** | Unmarried | 28 (76) | 63 (72) |
| | Married | 13 (31) | 24 (27) |
| | Refused to answer | 1 (2) | 1 (1) |
| **Current living arrangement** | Living alone | 2 (5) | 3 (3) |
| | With peer | – | 1 (1) |
| | Joint family | 25 (60) | 66 (75) |
| | Nuclear family | 15 (36) | 17 (19) |
| | Refused to answer | – | 1 (1) |
| **Occupation** | Unemployed | 4 (10) | 2 (2) |
| | Student | 9 (21) | 16 (18) |
| | Professional (Doctor, IT, Artist) | 9 (21) | 14 (16) |
| | Salaried | 12 (45) | 25 (28) |
| | Business | 4 (10) | 7 (8) |
| | Semi-skilled (driver, farmer, hotel, milkman) | 4 (10) | 4 (5) |
| | Skilled (lab techcian marketing, working with NGO) | – | 5 (6) |
| | Unskilled (shopkeeper, seller, labourer) | | 11 (13) |
| | Sex work | – | 2 (2) |
| | Refused to answer | – | 2 (2) |

**Table 2. Socio-demographic information of KII participants.**

| Characteristics | Response | KII N = 40 (%) |
|---|---|---|
| **Age** | Mean (SD) | 36.5 (10.68) |
| **Category** | Key MSM | 10 (25) |
| | NGO representative | 10 (25) |
| | Programme personnel | 9 (23) |
| | Counsellor | 6 (15) |
| | Doctor | 5 (13) |
| **Residence** | Urban | 20 (50) |
| | Rural | 20 (50) |
| **Religion** | Hindu | 30 (75) |
| | Muslim | 3 (8) |
| | Other (Sikh) | 1 (3) |
| | Refused to answer | 6 (15) |
| **Highest education completed** | High school | 4 (10) |
| | Higher secondary | 2 (5) |
| | Diploma | 1 (3) |
| | Graduate | 18 (45) |
| | Post graduate and above | 14 (35) |
| | Refused to answer | 1 (3) |
| **Marital status** | Unmarried | 16 (40) |
| | Married | 20 (50) |
| | Divorced/separated | 2 (5) |
| | Refused to answer | 2 (5) |
| **Current living arrangement** | With peer | 1 (3) |
| | Joint family | 18 (45) |
| | Nuclear family | 19 (48) |
| | Refused to answer | 2 (5) |

*"My first sexual relationship happened with my cousin. I mean to say that [/there is/] no need of apps. Normally you can find people around you."* [##OR-U-FGD-04-R5##]

Closed group networking among MSM indicated their need to prevent violence and being abandoned by their families and it made them fiercely protective of their identity. Stigma and violence compel MSM to remain hidden, for example, they endure public insults, they are called names such as *'Geela'(Wet); 'Gud' (Jaggery); 'Hijra'(slang used for transgender people)*, '*Chakka' (slang used for transgender people)* which are of common occurrence. Consequent isolation and withdrawal catalyse their need to go in hiding and isolation. If they brave the world by 'coming out'; they face repercussions both at work and society.

*"Mostly they cannot accept themselves, their families de-motivate them… So,when they step out of their family, society does not accept them. They face problem when they go out for work, if someone knows what they are then it is like that he will get sexually abused, he will get physically abused, all these discriminations happen with them."* [##MP-U-KII-03##, programme personnel]

In addition, they are afraid of anticipated violence and therefore they remain hidden.

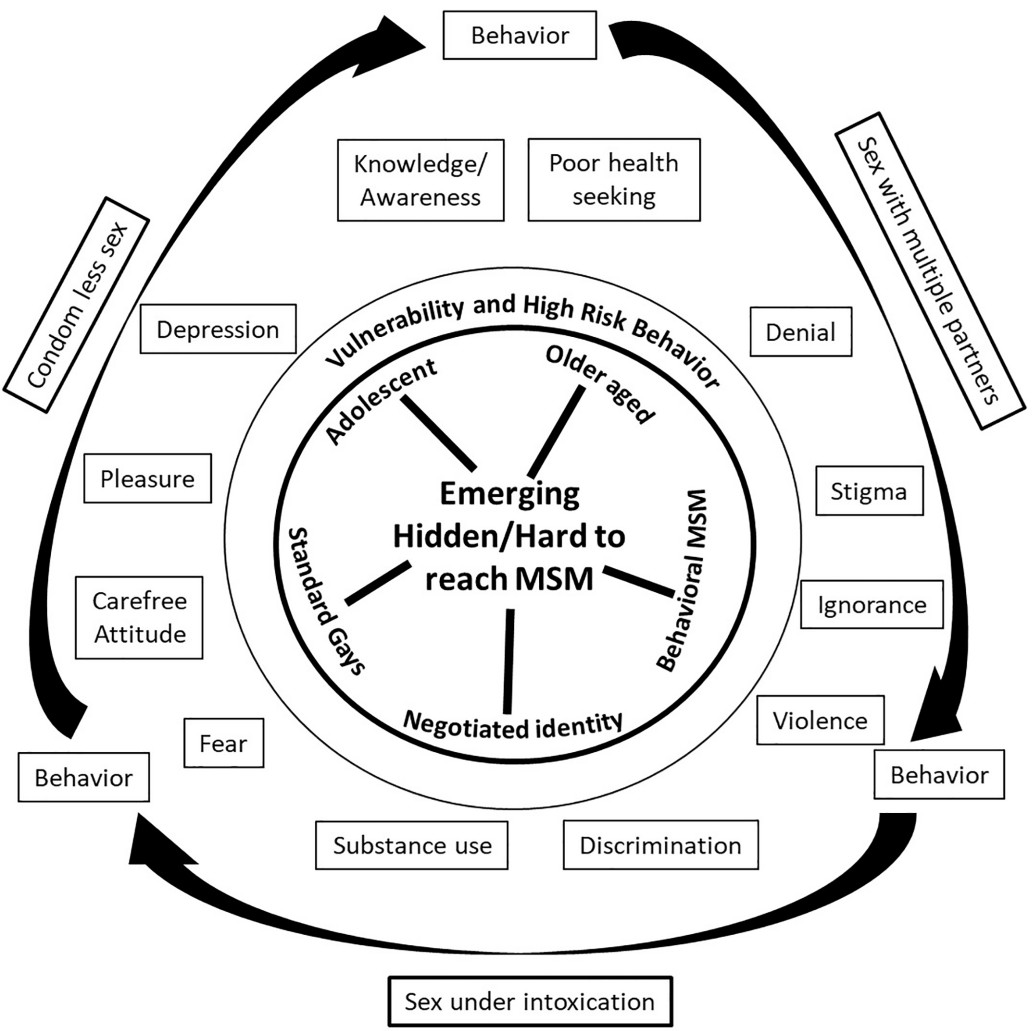

**Fig 1. Emerging Hidden/Hard to reach MSM and their vulnerability and high risk behavior.**

*"Whenever family comes to know about this [/sexual orientation/] they [/family/] abuse physically…"*
[##MP-U- FGD-01-R4##]

In the heteronormative work environment, repercussions of not remaining hidden is scary and riskier as seen in the experience narrated below:

*"I have two Kothi friends. They are so educated and from a learned family background. But one is in 'Khandra'... in prostitution. Yes, she [/preferred pronoun/] is HIV positive. But he [/she/] was well educated, belonged to a very good high-class family. He [/she/] was intelligent. But he [/she/] left everything and now he [/she/] is in prostitution… For money. We don't get jobs. We are harassed; we are tortured."* [##MH-R-FGD-02-R3##]

The syndetic of individual, societal, internalised and experiential stigmas lead to isolation/ withdrawal posing barrier to sex education, the emerging theme of 'Condom-less sex in hidden MSM' describes usage of condom among hidden MSM.

## Condom-less sex in hidden MSM

Condom-less sex was prevalent among hidden MSM. Impulsivity, lack of knowledge and carefree attitude towards condom use among hidden MSM contributed more to condom-less sex in all geographic regions.

*"If we have a condom then [/we/] use that, if there is no condom, then do not use it."* [##UP-R-IDI-04##, kothi, 27yrs]

*"If they find some good panthi, with a good body or good penis, then there are chances that people do it without condom."* [##MH-U-IDI-01_F##, double decker, 20yrs]

Likewise in other state like Odisha, in a rural setting, both fear of getting recognised and casual attitude towards condom use was evident when shared by a hidden MSM:

*"If I will go to buy condom to shop, then my parents will know that I am doing these types of activity. Like that, always I have fear. So, when, I do sex with my close friend, I do not use condom but when unknown person comes for sex, sometimes I use condom and sometimes I do not use. Now also I do sex without condom with temporary partners for sexual pleasure."* [##OR-R-IDI-08##, double decker, age not reported]

Another group of MSM is that of 'older MSM' who are transitioning into behavioural disinhibition. They are also becoming 'invisible MSM'.

*"Even now we have seen within our own circles that there are lot of people who were [/HIV/] negative when they were 30-40, who consistently used condom but now they become 50 years old, 55 years old and now they are like- I have lived my life, why do I need to use condom now? And these are ones who have started to test positive."* [##MH-U-KII-03##, NGO representative]

An FGD participant pointed out that MSM community is financially strong and influential, inducing younger hidden MSM who pursue riskier practices for entertainment such as chemsex contributing riskier practices and condom-less sex.

*"They consume intoxicating substances. Then many sex partners will do with one bottom. They are not in a mental state to use condom. It has caused day by day increase in the number of HIV cases in these people. They are financially strong, they are educated, they possess good positions in good companies, but, they are positive. And their number is increasing."* [##MH-U-FGD-02-R7##]

Unprotected sex leads to the fear of acquiring HIV and MSM get into denial mode. Despite existing HIV testing facility, hidden MSM avoided HIV testing and shared irrational fear of HIV diagnosis/ treatment.

*"Biggest reason is they do it without condom sex, so they don't go for testing. Because they are afraid of result… One of my partners says, it is better to die this way rather than knowing about result."* [##MH-R-KII-05##, NGO representative]

*"There are few MSM who are not afraid of getting tested, but they are afraid of the thought that if the result is positive, he will have to take lifelong treatment for that."* [##MH-U-FGD-02-R7##]

It also seemed that different typologies of MSM may negotiate condoms differently and therefore, identifying typologies is important.

*"right… I mean generally gays buy condoms but straight and bisexual guys feel very ashamed in buying condoms from the shop."* [##MP-U-KII-07##, key MSM]

To implement any intervention, the target population needs to be reached. It is critical to identify new categories that are currently not visible or identified.

## Expanding demographics of hidden MSM

All the hidden MSM respondents in this study had not accessed the existing HIV testing services. It is critical to reach 'hidden MSM' for prevention and control. The hidden MSM were the 'new MSM' who could be identified in this study and classified by their: demography, personality, and social status. All these new MSM were ignorant of risk and services and they were in denial mode.

*"Yes. Yes. So, these are the [/hidden/] populations. So, now is it clear little? So, I think I spoke about four populations so far, I spoke about: 1) A very young population, 2) Population in sex work that does not identify as gay, 3) Older population and 4) people who are in heterosexual relationships as well."* [##MH-U-KII-03##, NGO representative]

Box 1 summarises the emerging new MSM who remain hidden. The salient features are depicted and results are described subsequently.Box 1.–Emerging new hidden MSM and their salient features.

| S. No. | Emerging new hidden MSM | Salient features |
|---|---|---|
| 1. | Adolescent & Young | Features:<br>Young age: (7 years onwards)<br>• Sexually abused by adults<br>• Consensual sex between peers<br>Trigger:<br>Ignorance and curiosity, playful activities, and violent coercive encounters leading to habit formation<br>Perceptions:<br>Perceive homosexuality as norm |
| 2. | Sugar daddies- the older MSM | Features:<br>50 years and above who slip from continuum of programmatic prevention umbrella with risky sexual experimentations, coerce & sexually abuse very young boys; induce younger men, and they are financially well-off<br>Trigger:<br>Boredom, burn out, denial of self-risk, felt need to experiment, unrepentant about risk to younger groups |
| 3. | Standard gays: MSM in denial | Features:<br>Elite, well dressed, and very presentable<br>Status:<br>Financially well off<br>Trigger:<br>Practices male to male sex and considers this being his own norm<br>Characteristics:<br>Vehement denial of being an MSM<br>Socially:<br>Strict and keep to themselves and resistant to the MSM community group |
| 4. | Behavioral MSM | Features:<br>Professionals trying to comply with the clients' behaviors<br>Trigger:<br>To be part of client's activities in profession to please the client<br>Risk:<br>Ignorant that they are MSM |

| S. No. | Emerging new hidden MSM | Salient features |
|---|---|---|
| 5. | MSM with Negotiated Identities | Features:<br>Married or plan to be married<br>Trigger:<br>Fear and stigma<br>Characteristics:<br>Hidden and living in dual identity<br>Socially:<br>Conform to the heteronormative norms |

***Demography 1: Adolescent & young.*** The young adolescent MSM emerged as new MSM who were unaware of the risks and did not have clue to any existing services. In all states, at both urban and rural settings, the initial sexual abuse perpetrated by man towards male child when he was studying in 2<sup>nd</sup> standard to 6<sup>th</sup> standard (7 years onwards) was observed.

*"I do since childhood, No. Haan [/yes/], since 5<sup>th</sup> or 6<sup>th</sup> [/std/]."* [##MH-U-IDI-01##, versatile, student, 20yrs]

*"ahh… I had a partner when I was in 5<sup>th</sup> or 6<sup>th</sup> standard. He was my friend. …he was also my partner, he was my sex partner."* [##MP-U-IDI-08##, bisexual, 23yrs]

Getting into sexual relationship was many a times, of their own free will, especially among adolescents. Another respondent from Odisha shared his preference for sexual relationship changing from girls to boys during his adolescence days:

*"Before knowing about gay person I had sex with girls usually and my first sex hook-up was with my girlfriend's brother. So, it was like a new experience. Then I got to know that I am not that much comfortable with girls. I was in class 8<sup>th</sup> or 9<sup>th</sup> that time."* [##OR-U-FGD-04-R4##]

These respondents, now adults shared retrospectively about being sexually abused by men at a very young age which was painful. A hidden adult MSM from rural Maharashtra explained how he had the first sexual encounter with another boy in a playful setting which was painful but he kept reverting back to it; finding pleasure in it subsequently. They reported sexually experimenting with other boys in the meadows in rural settings, hostels in school settings or unsupervised homes out of school hours.

*"Means, initially I didn't know anything about this, means in young age, nothing will be understood, isn't it?... In village, we were going to roam in all the farms etc., and I also used to touch bodies of other boys intentionally. Means, I had that habit that time. It [/sexual contact/] happened while playing only. It was paining, means there was negative feeling for that [/sex/] and after… even if [/I/] said 'No' today [/that day/] but then also after 7-8 days, I used to go on the same path on that way [/have sex/]."* [##MH-R-IDI-02##, kothi, 20yrs]

Sexual explorations leading to male to male sexual activity was prevalent in other states also.

*"and… in my home town, in =Andhra Pradesh= Ummm… there I had [/sexual contact/] and that was young age. I think all, everyone explores. So, they were my age group people who would all be exploring their sexuality. So, it never struck that umm… ohh… you are doing something wrong, why are you doing this?"* [##MH-U-KII-01##, MSM leader]

Myths prevalent among young boys showed the need for appropriate sex education in schools.

*"Kids from 7ᵗʰ or 8ᵗʰ class [/came repeatedly to the stall set up for sex education at an exhibition in a village/]. So, we [KI] asked them, why do you want [/information/]?... Said, "Sir, we [kids] are means ummm... hostelites and in hostel, we do all these activities there… It got revealed that, means they… in general also, it [/Boy to boy sexual activity/] was in practice among many of the students. So, we [KI] asked them, why… means why do you do like this? Then what they [kids] replied was so shocking. They said that we have no idea that male-female sex happens, means our information is only that male-male sex is there."* [##MH-R-KII-01##, programme personnel]

*"I was getting attracted to boys. Everyone had a girlfriend then, so did I. when I came to 2nd year 12th, I met with a boy in tuition. May be he is no more. He asked me why I was staring at him all the time, what's the matter. I said may be there is something? Then he called me and said, "I think you love me."* [##OR-U-KI-07##, a highly educated corporate adult MSM]

Past sexual abuse suffered at very young ages was shared by the respondents. One cannot say whether it was orientation or if a habit got formed as result of abuse meted out.

*"At that time my age was at least 14–15 years. Understand, 14 years. Even before this, sex had happened. Some sex is painful… Painful people... Bleeding... It hurts a lot. This sex happened to me at the age of 14-15 years. I kept on refusing... I refused - anal sex was done."* [##UP-R-IDI-03##, kothi, 27yrs]

Evidence kept emerging further showing the preference of older MSM who reportedly exploited adolescent boys. The narratives under this theme gives insights into the exploitations and abuse of younger boys.

***Demography 2: Sugar daddies/ older MSM.*** Existence of 'sugar daddies/ older MSM' emerged in all states. MSM who were older, 50 and above aged, seemed to slip from the continuum of programmatic prevention umbrella reportedly out of boredom, burn out, denial of self-risk, and they were openly exploiting younger boys and men.

*"Anal sex… when I was in 9ᵗʰ std… I was young, and many [/older/] people like young. So, I used to fit in their criteria…"* [##MH-R-KII-05##, NGO representative, kothi]

*"Usually, I do not like such people, I like young people… some like very young people, some people even like minors."* [##MP-R-IDI-04##, panthi, 21yrs]

These older and visible MSM need renewed attention from programme as they got registered ages ago. The key informant who talked about older MSM also pointed towards social, probably moralistic, assumptions about no sexual activities among older people.

*"Another population that I feel is extremely vulnerable but is out of the purview of TI, is older [/MSM/] population and this is because we assume that after the age of 45-50 yrs., gay men do not have sex. This is unfortunately or fortunately not true because everyone needs sex. So, this population in particular what they will tend to do is, they'll hire male sex workers, or they will have sex with people who are 22-23 [/yrs/]. These older men are sexually very active [/and/] because they have money, they can totally be in the role of "Sugar Daddy" you know where they will get a young person, access to nice hot parties or get them alcohol. But no one reaches to this population [/older population/]."* [##MH-U-KII-03##, NGO representative]

From the young MSM's point of view, a belief emerged that confidentiality would get protected if they were having sex with older MSM. Another key informant from Madhya Pradesh shared:

*"So, the thing is they [/young MSM/] think that if we are having sex, then it's better to do it [/sex/] with old age people so that our identity remains hidden, and no one will get to know."* [##MP-U-KII-01##, NGO representative]

*"The youngest of our group is 16-17 years old, and one thing is that as many people go to the site, the more people who are young at the same time are preferred more by them [/Giriya/]. Giriya are willing to spend the money."* [##UP-U-IDI-01##, kothi 48yrs]

This is a call for reaching older men but not only for HIV prevention but they must be looked at from human rights frameworks and child protection act in the country. They are hidden even if once upon a time they were registered with the programme.

*"... =Asu=, a prostitute, …Now they [/preferred pronoun/] are old currently, they are at least 50-55 years old, but right now they want only young boys by searching (laughing)... 17-18 years old, just don't like old age..."* [##UP-U-IDI-03##, kothi, 20yrs]

The HIV epidemic would sustain if younger men/ boys are continued to be victimised and abused as we witness the transformation of 'visible' MSM to 'invisible' 'older MSM' who do not use protection culminating into 'sugar daddies' for their young victims. The darker hot spots should be supervised to prevent any nefarious activity. Narratives are shared from all the states:

*"There is one more place, =name of famous auditorium= for 'uncles' only. Whatever performance is going on, they go there to find young boys. Yes, the function goes on at the front, and these affairs go in the rear. Whatever that can happen in the back seats does happen. After that if they take those boys home or any other place. I don't know much about money. May be they take money."* [##OR-U-KII-07##, gay versatile, highly educated MSM]

Easy money and easy prey seemed to be driving the network dynamics of younger and older MSM.

**Demography 3: 'Standard gays- MSM in denial'.** Community defined the most elite of the MSM as 'Standard Gay' who never felt that they were MSM and therefore, remain hidden by default. Therefore, unlike older MSM/ sugar daddies, 'standard gays' are unique as they are in vehement denial of being an MSM. They are the most resistant ones.

*"… standard gays [/upper class/]."* [##MP-R-KII-06##, programme personnel]

*"…hidden MSM… in that category there are the ones which we call 'standard gays' and the people who consider themselves highly qualified… there are some people who keep things to themselves like [/they would say/] yes I am what I am. I mean they are like, I don't belong to the [/MSM/] community. They believe that they are not from the community… mostly you will see that the 'standard gays' will come in this category. He looks by his body features or his dress that he is normal. I mean, he wears branded clothes like 'Peter England's' but from their body features or their style we can get to know that they are from the community. When you gradually talk to them, you can figure out that they are from the community but they don't want to come out from that area [/Non MSM population/], they want to live in that zone [/their own zone of Cis/]. [/They say/] I am okay, I am happy and whatever I am, I am good…"* [##MP-R-KII-09##, Counsellor]

There are many other personnel who do not want to be identified and remain hidden.

*"…hmm mostly students, ma'am student who are in college, they do not want to open their identity in front of other students, I mean they keep themselves hidden…Top level government official, politician etc. hide their identity."* [##MP-R-KII-10##, programme personnel]

***Demography 4: Behavioural MSM.*** There were MSM who were ignorant, i.e., they do not realise that they could come under the umbrella of MSM.

*"ahh. There are a lot of people, according to them, sleeping with men does not make you gay or bisexual [starts laughing]."* [##MP-U-FGD-01-R3##]

A respondent shared about financial needs to be met through men getting into sex work; these men who accept/ practice same sex behaviour but they do not identify themselves as MSM' and therefore remain hidden.

*"Then there are some communities who don't identify as being gay at all but are into I would say male to male sex like for example male sex workers. Like in =Bombay= you have a substantial population that comes from =name of state=, = name of state =, they have to act in films so they come with film stars' hobbies. They'll come to =Bombay= and rent being expensive, portfolio being expensive; they'll get into sex work to keep things going. So, this population does not even identify as gay. So, this population is again a very difficult population to tap."* [##MH-U-IDI-03##, panthi, 40yrs]

We observed that some of the men showed adoption of MSM behaviours due to the need or desire of their professional clients. These type of MSM are the 'Behavioural MSM' who might not have homosexual orientation of their own and therefore remain hidden. They lack the knowledge of safer behaviours, preventions and available services. They believe that they are not MSM or gay. So where is the risk again? There would be no felt need to access TI meant for 'MSM'

*"Because a lot many models, gym trainers who are not actually homosexuals or bisexuals but they are doing it because they have their profession in them… So, you can find those [/hidden/] MSM."* [##MH-U-KII-04##, doctor]

***Demography 5: Negotiated identities.*** Owing to prevalent stigma and complete non-acceptance, MSM try to conform to the heteronormative norms by getting married, or align with their biological gender; suppress their feelings or keep their identity hidden. Thus, this pressure keeps them away from accessing services meant for MSM for fear of being recognised.

*"…both are hidden but mostly the married ones, because they do not disclose their identity…"* [##MP-U-KII-04##, programme personnel]

*"… now at that time, nobody has a fear from the society, all are scared from their family."* [##MP-U-IDI-08##, bisexual, 23yrs]

Inclusivity and acceptability for sexual minorities have been emphasized at the primary level itself as it might normalise diverse identities at younger age creating safer and affirming environment.

*"First thing, this can be studied in book that they are as normal as other people and we need to accept them. There is nothing difficult or different about that."* [##OR-U-KII-03##, key MSM]

## Reaching the hidden MSM

The need to reach the hidden MSM became apparent and the major concern was where and how to reach this vulnerable elusive population which is discussed further. Three major facilitators to reach the hidden MSM emerged as: (a) Ambient policies, (b) Venues/ spots, and (c) Peer Engagement.

### a. Ambient policies

Legal and social protections would help them to come out of the closet and possibly access the services; hence the next emerging theme was the 'Ambient policies'. To reach hidden MSM, government support was an expectation:

> *"Why will they identify themselves? What will they get? Human beings are selfish. If they had any legal and social protection, they would have certainly identified themselves."* [##OR-R-KII-06##, MSM leader]

The same respondent made suggestions for due recognition of MSM population within the legal system. When law affirms diverse identities, it can send strong message for safety and protection of the community. Supportive legal environment would facilitate 'coming out' without fear.

> *"There must be a formation of policy for legal protection. The way these things have been done for TGs, same way it should have been done for the source as well, i.e., the MSM. There should be a mention of gays in the NALSA judgement [/ Judgment recognizing transgender individuals as the 'third gender/]. If you go to western =Odisha= you will find MSM dressed in sarees. They live as gays."* [##OR-U-KII-06_F##, MSM leader]

A lack of political support renders MSM into hiding imposing barriers to health care access. Helpline numbers can be useful as support. Helplines might prove to be safe space for the hidden population and it may later bring in trustworthy interactions to reach this hidden population:

> *"Here so many politicians and ministers, they do not support MSM. They are saying that gay culture is not allowed in India. That's why MSM are not getting any social support. For this they are hiding themselves… Through government you can keep one helpline number for MSM. MSM can call in that number. And they can message you."* [##OR-U-FGD-04-R2##]

### b. Venues/ hot spots

The congregation spots and isolated MSM frequented spots need to be mapped continuously to reach new population. The venues, where they are able to operate freely without threat of being recognised, should be monitored to reach this population. Several venues were identified as hot spots in all the sites and they were: malls, gardens, railway bridges, religious fairs and several well-known residential localities. Trains and railway stations emerged as important hot spot for reaching *'panthis' [/male usually macho MSM/]*.

> *"In =name of long distance express train= we get good ones. We get so many 'panthis'."* [##MH-U-FGD-01-R6##]

To maintain their privacy, MSM also go to tourist destinations where they become part of the 'anonymous' crowd and have freedom to express themselves without any scrutiny:

> *"We used to book rooms for us [/at Goa-a tourist place in India/]. And those of us who are held back here in our city [/ remain hidden/] we used to let ourselves go. We used to shave our facial hair and dress up in skirts and dresses. So, guys used to approach us. Those who were interested in this, did it, those who were not, they used to sit in the room."* [##MH-R-IDI-01##, kothi, 25yrs]

In Odisha, hidden MSM are difficult to identify and they can only be visible to an MSM at the hot spot.

*"We have no such group of MSM because we are staying in a society… But in society we are staying like a male [/ straight/] person. Always in evening time we go to our fixed place where always we meet… In evening, usually we go to the river bank and there I find them. We do not have a big network."* [##OR-R-IDI-01##, kothi, 19yrs]

At MP states hot spots were identified:

*"=Narmada River's =name of ghat="* while in UP state *"=name of talkies=, rural area like =DEVA=, =Shahdatganj=, next =Shabdarganj=, =Zaidpur=, =Jhrnaa=is near to =Barail=…"* [##MP-R-KII-05##, key MSM; ##UP-R-KII-05##, doctor]

In this study, MSM were invisible because of the sensitivities and the trust issues. This sensitive hidden population would only become visible to those in whom they had trust or got interested. The former groups were mainly that of peers and hence 'Peer engagement' was the emerging theme for reaching this population.

### c. Peer engagement

Public acknowledgement of membership into MSM community seemed very threatening. Since 'hidden' MSM are hidden because they want to remain hidden, they fiercely protect their identity. It is a challenge to reach them as stated by most of the participants. Many of them appear straight and it is almost impossible to identify them.

*"Very hard… when we [/MSM/] cannot reach them! Then how can you? Do you know, someone stays next to your home, but till the time he sends the photo, you don't know anything? And the one who stays nearby will not share photo. Because his identity will be disclosed. Therefore, they are called hidden."* [##MH-U-FGD-01-R1##]

Presence of MSM on social media like WhatsApp, Grindr etc. emerged. However, these are the platforms where MSM convene for fun and sexual activity and one may reach the 'visible' MSM. Contrary to this, respondents stated that to reach 'hidden' MSM through social media for health would be a challenge. A key informant informs about formation of 'homodes' which is a sex group in which interest in health care would be very limited.

*"I create a WhatsApp group and I add my 10 people in that group. Now these 10 people will then add 10 more of their people, those 10 people will add 10 more of their people and then it becomes a little universe 'homodes' [/A hyperactive spree in which your activities bother everyone except for you/] of 250 people. And there are infinite number of homodes on WhatsApp and all of these become good functional sex groups potentially because all people need to do is put their photographs, put their videos, talk about their sexual desires… So, you know all of those communities are extremely hidden, none of them particularly, really need to access community-based centre [/TI/] because all they are connecting probably for sex… if you have to focus on the populations that are hidden within the hidden community, a place like WhatsApp is very very difficult to penetrate."* [##MH-U-KII-03##, NGO representative]

Another FGD participant confirmed:

*"Those who are hidden, they will not pass phone numbers… all those who use social apps are open [/visible/]."* [##MH-U-FGD-01-R1##]

Peers emerged as the most acceptable persons to reach 'hidden' MSM.

*"Mostly I prefer that their staff…gays should be there as staff…in these departments…actually what happens is that gays cannot open in front of straight people as they can open to gay people or trans genders or lesbians means… So, they will feel more emotionally touched and comfortable also."* [##MP-R-KII-09##, counsellor]

The importance of networks to reach the hidden MSM were cited. Peers appeared to be strong contenders to reach hidden MSM. A carefully selected peer may become a strong influencer in his own network.

> *"To find out a hidden person… to understand this… only the hidden MSM can find out another hidden MSM."* [##MH-U-FGD-01-R2##]

It was felt that peer approach may be a successful in reaching hidden MSM especially '*Panthis/ Giriya'/* Standard/ Behavioral MSM who live like straight people; thus, difficult to identify as they themselves also do not self-identify as MSM. Peers can naturally explain and understand thus facilitating their reach to the health care.

> *"Mostly MSM do not accept that they are gay or bisexual."* [##OR-U-KII-07##, MSM leader]

> *"If they don't accept themselves as MSM then a person from the community can only explain them. It's a time taking process especially 'giriya' and double-decker never accept it easily... Mainly giriya wants to be hidden; so, he never accepts it easily."* [##UP-R-KII-01##, MSM leader]

### Needs

Three critical needs emerge from the data. To self-actualize, the respondents voiced first need to find a place in the world, they needed to belong to and accepted by others and thus: (a) 'counselling' emerged as a critical need.

#### a. Affirming counselling needs of hidden MSM

> *"I just want to say that those people who are hidden now, we need to provide them counselling so they can come forward freely."* [##MP-U-KII-03##, programme personnel]

Loneliness, depression and suicidal ideation were reported among hidden MSM in both urban and rural settings.

> *"As he [/an MSM/] does not get this [/family/] support, he will not share his problems anywhere… then internally he is… what they say… he gets depressed. He always feels alone."* [##MH-U-FGD-02-R7##]

> *"Then, some are married. After their marriage breaks, we feel that we are doing some wrong things. Some don't have kids. Somebody's wife has eloped.…It is a mental discomfort. If he is alone at home, he will hang himself. People have died like this."* [##MH-R-FGD-02-R2##,]

> *"I can survive my life but how can I live alone."* [##MP-U-IDI-08##, bisexual, 23yrs]

The community also cautioned that qualified personnel from the community are needed who are well trained to be non-judgmental and empathetic.

> *"…there should be some good counsellor mainly who belongs from the community, the one who should have done MSW [/Masters in Social Work/] because he [/counsellor/] can understand the community very well."* [##MP-U-KII-01##, NGO representative]

'Need to talk' meant a place to interact, along with a health care provider to talk with, kept emerging throughout the narratives in one or another form. Affirming to the MSM behaviours by the health care providers was an expectation.

*"When we go to hospital for check-up, doctor should behave as a friend and should not be in hurry, talk politely and make us comfortable."* [##OR-R-IDI-02##, kothi, 22yrs]

Ambient environment and affirmative communication at the government facility needs improvement.

*"Yeah. That was the first time that's why maybe I don't want to go in any government hospital. In private hospital, they behave normally, the way they talk… they know how to talk with the people."* [##MP-U-IDI-08##, bisexual, 23yrs]

Reaction of a health care provider proved to be a poor non-verbal communication example cited by a respondent:

*"…if I tell to a straight doctor that I am gay and I have done sex with a boy; so somewhere they feel bit different, they stay silent, they don't tell anything or they are not able to say anything."* [##MP-U-IDI-08##, bisexual, 23yrs]

In context of these mental health issues, skilful communication is required for MSM. Skilful communication involves not just conveying information, but also ensuring that the message is received, understood, and engages the other person effectively.

*"If you go and talk with this population about HIV, they will tell I am not at risk at all. But on the other hand, if you go for 'How are you feeling today'? Tell me what happened, then there is a possibility that this population will talk to you."* [##MH-U-KII-03##, NGO representative]

Education interventions in schools especially for boys might be required for making informed choices for inclusivity, acceptance of diversity, healthy relationship and sexuality.

*"All the information regarding HIV and safe sex should be given as a part of education. If it is in syllabus since child level and as the child grows up and his age increases, as our physical growth takes place, this knowledge will be given accordingly. Then he will do safe sex."* [##MH-U-FGD-02-R7##]

*"Some words are not known. Penis, semen, vagina etc. are cultured… Sanskritised words… The language should be public language, or it should be explained by teachers or counsellors."* [##MH-U-FGD-02-R2##]

MSM in this study showed awareness of need for HIV testing but their major concern was internalisation of social stigma and conflict of gender roles and norms in terms of gender-nonconformity, stigma, violence and secrecy that made them, lonely, anxious and depressed. Hence the second emerging need was: (b) MSM focused facility for HIV test:

### b. MSM focused facility for HIV test

The process and the associated logistics pertaining to HIV testing at the facility inhibits MSM to remain hidden and avoid getting tested.

*"Why do they choose to remain hidden? Right? That's what, ma'am here confidentiality is a very big factor. If they don't perceive that their information will be confidential… so if need to wait in line somewhere or if results may get exchanged [/breached/]. These are as it is population who are not very enthusiastic about getting an HIV test. So, just the entire logistical aspect of it also makes them not want to come in and test."* [##MH-U-KII-03##, NGO representative]

Changing the ambience at TI was an expectation at all four states.

*"Because TI, condition of TI clinics are not very good. So, those MSM who are actually not from a very good socio-economic backgrounds, they will not feel any problem in going to TIs. But, those from a better and average and more than, above average socio- economic background [/hidden/], they will never go to TIs, this is a problem with this".* [##MH-U-KII-04##, doctor]

Informal light atmosphere with recreation arrangements and safer environment would enhance acceptance and usage of the facility especially for HIV testing. Both respondents from Uttar Pradesh and Maharashtra asked for hybrid specified space with small recreation and HIV testing both. Importance of recreational activity at HIV testing centre was emphasized by all sites.

*"…so if there is a community centre which has a small coffee shop in it, people will come to access and when all these, there are so many like library, internet and umm... things and plus you have these HIV umm… section and then… section for a…STD."* [##MH-U-KII-01##, MSM leader]

*"I just want… even every MSM guy wants that there should be spots for them, so that they can do whatever they want easily."* [##MP-U-IDI-03##, bisexual, 22yrs]

*"Without light atmosphere, for MSM, there would be eerie anticipation of HIV results… but if they go to the ICTC, there is only counselling… no activity [/music etc/] is seen, then people are afraid of that, there is a little fear that right now you will come out! Now, you will come out! Now, you will be done! [/HIV positive/]."* [##UP-R-KII-05##, MSM leader]

## Discussion

India has a strong HIV prevention and control programme but, their potential effectiveness is limited by structural factors that contribute to poor health-seeking behaviours among MSM. This study brings evidence about the existence of hidden MSM, who were young, unravelling the intersectionality of child sexual abuse and HIV high risk behaviours of older MSM. To create an effective response to HIV prevention and control, programme needs to take strong position against sexual violence and exploitation against children empowering them with life skills for safety, learn about boundaries of protection and knowledge of where/ how to seek help.

Unlike other studies [13,14], this qualitative study shows that social media might not be the best option to reach hidden population for interventions. In case of hidden MSM, reaching them on social media was voiced as difficult. Peers were acceptable especially young MSM who were vulnerable due to multiple intersectionality of being young, ignorant, afraid and stigmatised. Almost all of them were afraid and had experienced violence and stigma and faced mental health problems. Reaching this hidden population is an imperative to reach the last mile and combat HIV epidemic [15,16].

Our earlier published paper [11] focuses on health-seeking behaviour of MSM from rural India while this paper encompasses data of MSM from urban and rural settings. The focus of this analysis brings evidence of new emerging typologies of MSM who are presently hidden. As shown in the analytical framework (Fig 1), these emerging typologies of MSM are currently getting missed from the programme definition. 'Denial, ignorance or failure to realize one's orientation' play a vital role in missing out MSM. This paper also highlights need to embed education on diverse populations at a broader structural level to bring inclusivity at primary level itself.

The emerging theme describes new demographics of MSM in India who are hidden viz. 1) Adolescent and young MSM, 2) Sugar daddies- the Older MSM, 3) Standard gays: MSM in denial, 4) Behavioural MSM and 5) MSM with negotiated identity. We report contextual condom-less sex among unreached hidden MSM which might be one of the key drivers of sustained epidemic observed in this population. As declared in NACP IV [17] guiding principles for universal access – inclusion of all high-risk MSM – regardless of sexual identity, marital status, age, or presumed/stated sexual practices

(receptive or penetrative or both) need to be included, and therefore definition of MSM needs to be expanded for future interventions.

The study shows an emergence of critical themes that summarize the experiences of the hidden and complex vulnerabilities of all hidden MSM described above. The most concerning findings are the reported sexual abuse of boys and younger men by older MSM, viz. Sugar daddies. The hidden MSM, being unreached by the programmes, easily miss out on prevention education. As reported in several studies from India and elsewhere [18–22], our study confirms that sexual encounter happens at a very young age (6–7 years), which is typically coercive, non-consensual, and obviously with no protection. The sexual encounter recalled by the participants during their adolescence was mostly coercive or exploitative by adult male/s; sometimes consensual between peers as a playful pastime. The young adult males were exploited by older MSM through inducements of financial incentives or expensive recreations. All of these sexual events are high risk without protection, which has been known to significantly increase the risk for sexually transmitted infections (STIs) and future risky sexual behaviour [23,24]. According to a study conducted by the Ministry of Women and Child Development in India, out of the total 53.22% child respondents reporting sexual abuse, having faced one or more forms of sexual abuse that included severe and other forms. Among them, 52.94% of boys reported experiencing one or more forms of sexual abuse, a figure higher than that reported among girls, which stood at 47.06% [25]. Severe forms of sexual abuse were reported by 20.90% of whom more than half were boys. A recent report states the age-standardised (20 and older) prevalence of sexual violence against males as 13.6% [26]. Further, this report gives a global picture of the first experience of sexual violence among adolescents and young people that occurred before the age of 18 years for 67.3% of female and 71.9% of male survivors. Social media platforms and websites have been reported as the primary mode of accessing children in the state of Goa [27]. Additionally, adolescence is the period of social peer and romantic focus [28], and true to the demands of adolescence, either because of the past or any new sexual activity, sequelae of consensual sex starts. In our study, sexual experimentation and the influence of friends and peer groups have been cited by the respondents to lead to same sex behaviours in spaces such as meadows, fields, unsupervised homes, and school hostels. Many times, the same sex behaviours persist as practice and continue during adulthood. These behaviours among adolescents occur because adolescents want to conform with peers and they are more concerned about what they perceive as normative rather than what is unhealthy [29,30]. For example, the misperception [31] reported in this study by the hostel inmates reporting prevalent same-sex behaviours among them was the norm known to them and practiced by them. Gaps pertaining to sex education and safety measures emerge. Since adolescents are known for their tentative and episodic abandonment of their own value system for the sake of gaining group acceptance [32], providing information to adolescents about the risks is not enough to discourage them from engaging in the high-risk behaviour, indicating a need for a considerable shift in conceptualizing intervention models for this population. Social norms and peer normative behaviours are not included in the adolescent reproductive and sexual health programmes. The programme may use a 'Social Norms Theory' based intervention for early initiation of social norm modification among young boys to reduce the amount of social norm correction required at older ages. Social media such as Facebook or Instagram can be used for disseminated social norm corrective messages about delaying sexual debut and stopping unhealthy behaviours between young peer groups. Sexuality education needs to start at very young age of 7–8 years.

This study identifies a second group of hidden MSM. 'Older MSM/ sugar daddies' are the leakage in the programme continuum, showing burnout of prevention behaviours as they progress in age. The sugar daddies claimed to be experimenting and preferring younger boys for unprotected sexual activities through economic incentives. Therefore, these are the MSM who transition into 'Hidden within known MSM' (HWKM); technically, they are otherwise listed with the programme. It is a challenge to intervene with HWKM or older MSM because of complacency of the programme towards men's continuation of safer behaviours. As HWKM progress in age, their personality seemed to change and their behaviours became riskier. These are men who are older, hence their duration of risk increases [33], rendering them riskier than before. They also have financial resources to hire/ induce very young MSM, which adds to the already existing

risk for younger MSM, both boys and young adults, who need protection from sexual exploitation and abuse. There is a paucity of data on sugar daddy dynamics among MSM in India, and it needs to be examined for the possibility of behavioural disinhibition, which is a warning bell for the programme. Such behavioural disinhibitions have been reported among MSM in the context of the resurgence of syphilis and other STIs among MSM [34,35]. This resurgence was attributed to an abandonment of safer sex practices among some gay and bisexual men [36]. Additionally, their interest in young boys and men indicate a pro-offending attitudes or cognitive distortions. A re-examination of the registered older MSM who might not be practising safer behaviours now, needs to be taken up urgently for appropriate interventions to be implemented to respond to the need to protect younger boys/ men and also address the sustained HIV epidemic. However, the protection of young boys and men who are knowingly or unknowingly victims of sexual abuse and exploitation is of utmost importance. HWKM turns out to be a typical sex offender who tends to be manipulative. Designing psychological interventions such as cognitive behavioral therapies (CBT) for older HWKM, along with redeveloping safer sexual practices, developing corrective strategies and policies against sexual exploitation is the intersectionality that the programme may need to address. As recommended by others [37], we also recommend integrating counselling to reduce condom less anal sex for HWKM.

The younger age of new/ hidden adolescent MSM and their sexual interactions with the 'sugar daddies' in this study gives rise to an urgent need to reach hidden MSM who are mostly young; reported sexual encounters by adult men at a very early age, and sometimes consensual between peers. An ECPAT international report showed that the boys who were subjected to sexual exploitation were at risk of physical injury, mental health concerns and substance misuse, and HIV/AIDS and other forms of sexually transmitted infections [38]. Traditionally unimaginable to general parents, teachers, or guardians [39,40], data show that young boys are vulnerable [26]. Our study emphasizes the crucial and urgent need to protect young boys who are traditionally considered 'safe' by the families in India. Stigma in the HRMSM undermines health by preventing access to critical health-promoting resources, acting as a stressor leading to health inequities [41], indicating a need for stigma-free health services for young adolescent boys in India. Affirmative care from Health Care Provider (HCPs) at the facility would be required to provide care to the young adolescents.

Another hidden MSM group identified was the 'behavioural MSM' who might not be accessing prevention services as they would not identify themselves as MSM. Other studies have also reported 'straight identifying MSM' which conceptualised the lived experience of these men [42].

The 'MSM living with negotiated identity' also remain hidden, while living with families/ community, displaying biological gender conforming practices and behaviours. A study by Ekstrand et al., showed concordant report wherein men who identified themselves as 'bisexual' were not out to others, married, and presented themselves as straight persons in the community, and had unprotected anal sex [43]. Our participants also reported homosexual contacts among married MSM continued as they were unable to resist the need.

Finding hidden MSM becomes a challenge because of the spectrum of hidden 'adolescent MSM', elite 'standard MSM' in denial, the visible but currently invisible 'older MSM', and 'Married MSM living in negotiated identity'. Along with such a wide spectrum, there is the phenomenon of transitioning spaces and populations. Most of the 'hidden MSM' still went to the same physical hotspots, but they were also using social media for sexual networking while anonymising themselves. NACP V emphasized reaching the hidden population who are using various social media apps. The digital interventions have shown promise with MSM [44] who voluntarily opt for interventions. However, the hidden MSM are the resistant ones, reaching those needs tailored interventions and strategies. In-depth discussions and participant observations of various venues could be done with the help of peers. The presence of MSM on virtual platforms is well known and well documented. Social media usage is increasing among these groups across all education and income levels, facilitating MSM and TGW to find partners quickly, conveniently, and in larger numbers than in-person dating methods [45]. However, our study indicated social media to be popular for having fun, soliciting partners covertly, while fiercely protecting their identity. The present data showed that locating or finding 'hidden' MSM on social media would be a challenge. Most of the

MSM on social media who do not want to be recognised would remain hidden by having a fake identity. Although dating app users' privacy concerns are known to be focused on privacy and policy at an 'institutional' level, the users are more concerned about how the service would use their data [46], and therefore, 'coming out' to others by hidden MSM becomes a remote possibility except for their sexual partners. For example, many online hook-up sites allowed users the option to disclose their HIV/ PrEP status to potential partners, which was a strong intervention, but the majority of MSM did not use these features [47]. Thus, the online space allows MSM to find sexual partners without disclosing a lot of personal information. Additionally, it is easier to reach those who want to be reached but it becomes a humongous challenge to reach the elusive hidden population who resist contact with the outside world.

Most of the hidden MSM (IDI) and visible MSM (KII/ FGD) talked about the need for 'peer approach'. Peer referral model emerged as acceptable model; it promotes trust and social acceptance; it might be a successful option to reach 'hidden' MSM. The programme advocates the concept of 'community champions' [48], who could be the potential 'seed' for reaching the initial hidden MSM who may refer other network members without compromising confidentiality by encouraging autonomous voluntary service access. MSM-focused facility with affirmative care, peer model for referrals and counselling for mental health might facilitate access and service utilisation among hidden MSM.

Since some of the MSM in our study were reportedly married and also hidden, HIV testing of their spouses is a missed opportunity. In India, HIV testing for adults is conducted only after counselling and obtaining informed consent. However, facilitating HIV testing among adolescents presents a foreseeable challenge in India due to ethical challenges of administering consent to the minors. Countries like Myanmar, Sri Lanka, and Thailand have changed HIV policies to allow adolescents who demonstrate sufficient maturity and understanding to consent independently to HIV tests [49]. Literature does indicate that non-disclosure of sexual identity affects HIV service utilization [50–52], yet much remains unknown about the barriers faced by hidden MSM in India in accessing HIV services, hindering the development of culturally-relevant and responsive HIV prevention approaches. We recommend developing policies for HIV testing for adolescents with/ without parental consent, which would address the intersectionality of sexual risk and sexual violence for young males in India. The task may require strong advocacy at all levels of the social, legal, and political ecosystem.

## Limitations

The qualitative study was conducted in selected districts of the four states of the country, the data derived cannot be used to make generalization. However, the geographic variations in the districts add strength to the findings with respect to different scenarios observed in these geographically sparse and culturally different states. The findings can be extrapolated to states with culturally similar states with high and low HIV prevalence. Since, this was more of an exploratory study, it could inform programme on additional high-risk MSM categories that are not currently included under the definition of MSM of Targeted Interventions and strategies to reach them to link to programme. The data is highly relevant for programme today considering that these key population are driving the sustained epidemic currently. This study identified new hidden populations and focuses on issues pertaining to all HRMSM which feeds into the Strategy Document of National AIDS and STD Control Programme Phase-V target population (2025−26) [53] which recognises that many of the target population may not identify themselves as being at risk and have poor risk perception but they urgently need to be covered by the programme.

## Conclusions

Very young adolescent MSM were identified along with several new categories of hidden MSM who did not consider themselves as MSM. Peer approach for outreach was acceptable and feasible for reaching hidden MSM. Protection of children from child abuse calls for strong laws. Education of young boys in sex, sexuality and safer sex practices and empowering them about laws and how to protect themselves from sexual abuses emerges. There is an urgent need for an ambient environment for boys from primary level of schools to bring in inclusivity for diversity. The intersectionality of being MSM

and young male needs to be addressed by the HIV programme and education system. Awareness of young boys and adolescents through 'Adolescent Reproductive and Sexual Health Education (ARSHE)' is imperative. Protection of children call for the highest level of endeavour by taking strong actions against perpetrators and ensuring protection of male at family, societal, community and political level. At the same time families, parents and teachers need to be educated on preventing and reporting episodes of sexual abuse.

MSM especially beyond 50 years may go into behavioural disinhibition. Program must revisit MSM reaching middle age, i.e., the sugar daddies and uncles for sustaining safer behaviors, learning POCSO Act and preventing exploitations of younger boys and men by providing psychological counselling.

## Acknowledgments

The study was led by ICMR- National Institute of Translational Virology & AIDS Research (ICMR-NITVAR) [Formerly, ICMR-National AIDS Research Institute (ICMR-NARI)], Pune, India. We acknowledge the support received from Director, ICMR-NITVAR for the entire study. We are thankful for the continued support received by the Indian Council of Medical Research, India. We acknowledge the support received from National AIDS Control Organization, India right from the inception to the conclusion of Phase 1 of the study. It was critical to receive support from State AIDS Control Societies of the respective states which were facilitated by NACO. We thank Dr. Rewa Kohli, Dr. Archana Verma and Mrs Nayana Yenbhar for their contribution in the study. We thank the support extended by the leadership of Family Health International (FHI360) and the leadership of the LINKAGES project. The entire study team expresses heartfelt thanks to our study participants without whom this study would not have been completed. We acknowledge Savitribai Phule Pune University for the contribution of the research fellows.

## Author contributions

**Conceptualization:** Seema Sahay.

**Data curation:** Seema Sahay, Girish Rahane, Tuman Lal Katendra, Amarendra Mahapatra, Shivendra Kumar Singh, Sanjeev Kumar, Sampada Bangar.

**Formal analysis:** Seema Sahay, Sampada Bangar.

**Funding acquisition:** Seema Sahay.

**Investigation:** Seema Sahay, Amarendra Mahapatra, Shivendra Kumar Singh, Sanjeev Kumar, Sampada Bangar.

**Methodology:** Seema Sahay, Sampada Bangar.

**Project administration:** Seema Sahay, Amarendra Mahapatra, Shivendra Kumar Singh, Sanjeev Kumar, Sampada Bangar.

**Resources:** Seema Sahay, P.S. Saravanamurthy, Bitra George, Sampada Bangar.

**Software:** Seema Sahay, Sampada Bangar.

**Supervision:** Seema Sahay, Amarendra Mahapatra, Shivendra Kumar Singh, Sanjeev Kumar, Sampada Bangar.

**Validation:** Seema Sahay, Sampada Bangar.

**Visualization:** Seema Sahay, Sampada Bangar.

**Writing – original draft:** Seema Sahay, Dhammasagar Ujagare, Sampada Bangar.

**Writing – review & editing:** Seema Sahay, Dhammasagar Ujagare, Girish Rahane, Tuman Lal Katendra, Amarendra Mahapatra, Shivendra Kumar Singh, Sanjeev Kumar, Chinmoyee Das, Bhawani Singh Kushwaha, Vinita Verma, Rajiv Ranjan Tiwari, Nupur Mahajan, P.S. Saravanamurthy, Bitra George, Sampada Bangar.

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
