## [Decision Letter · Decision Letter 0]

24 Sep 2025

Dear Dr. Sahay,

Thank you for submitting your manuscript to PLOS ONE. After careful consideration, we feel that it has merit but does not fully meet PLOS ONE’s publication criteria as it currently stands. Therefore, we invite you to submit a revised version of the manuscript that addresses the points raised during the review process.

We look forward to receiving your revised manuscript.

Kind regards,

Juan Pablo Gutierrez

Academic Editor

PLOS ONE

Journal Requirements:

“This work was supported by the United States Agency for International Development (USAID) through FHI 360/Linkages [Grant Number: AID-OAA-A-14-00045]. Corresponding author received the funds.”

4. For studies involving third-party data, we encourage authors to share any data specific to their analyses that they can legally distribute. PLOS recognizes, however, that authors may be using third-party data they do not have the rights to share. When third-party data cannot be publicly shared, authors must provide all information necessary for interested researchers to apply to gain access to the data. (https://journals.plos.org/plosone/s/data-availability#loc-acceptable-data-access-restrictions)

**Additional Editor Comments:**

Please review and address the concerns raised by the reviewers, particularly reviewer 2, regarding the methodological approach and analysis reported.  Clarity on how this paper relates to the already published study, whether previous results have been previously reported, and how those results differ from this paper is needed.

Reviewers' comments:

Reviewer's Responses to Questions

**Comments to the Author**

1. Is the manuscript technically sound, and do the data support the conclusions?

Reviewer #1: Yes

Reviewer #2: Yes

2. Has the statistical analysis been performed appropriately and rigorously?

Reviewer #1: Yes

Reviewer #2: N/A

3. Have the authors made all data underlying the findings in their manuscript fully available?

Reviewer #1: Yes

Reviewer #2: Yes

4. Is the manuscript presented in an intelligible fashion and written in standard English?

Reviewer #1: Yes

Reviewer #2: Yes

Reviewer #1: The data gathered through the study is very relevant and useful for potential improvement in the implementation of programs targeting MSM, not only in India but also in other countries. Overall, I felt that the article is very long which to some extent makes sense given the quotes from participants and the very diverse categories that were included. Nevertheless, the focus can get lost given the amount of categories and information. If possible, I suggest making it more concrete but I don't think this is an obstacle to accepting the paper.

A table showing the characteristics of participants would be useful to get a better sense of who is represented in the study. Also, the text in the quotes from participants may be difficult to understand for some people who are not used to the literal transcript of interviews, I suggest to simplify it.

Reviewer #2: GENERAL COMMENTS ON THE ARTICLE

The article presents original research data and addresses a crucial and highly relevant topic for HIV prevention in India, which may be of great use in the global fight against HIV. The empirical evidence included in the manuscript is extensive, and its results are robust. However, there are both technical and conceptual elements that would benefit from further review.

ON TIMELINESS

Relevance: The authors could initially elaborate on the relevance of their data, considering that these qualitative data were collected six years ago (2019), before the COVID-19 pandemic, which significantly altered health services worldwide. It would be advisable for the authors to explain why the data remain relevant today.

ON THE OBJECTIVE OF THE ARTICLE

In the article it is stated: “This study was conducted among MSM and other stakeholders to understand the population which is hidden, their issues and their expectations on how to reach them comfortably; without any threat.”

Comment: In this sentence you outline the study’s objective; however, I believe that the article’s objective could be more specific. It would be worthwhile to review and adapt this objective so that it aligns more directly with the manuscript’s findings.

ANSWERING THE QUESTION: IS THE MANUSCRIPT WELL ORGANIZED AND WRITTEN CLEARLY ENOUGH TO BE ACCESSIBLE TO NON-SPECIALISTS?

No

LENGTH OF THE DOCUMENT AND LACK OF BALANCE IN SECTIONS

Comment: The article is unnecessarily long, with 11,000 words. The Introduction has 1.5 pages (564 words), the Methods 2 pages (530 words), the Results 23 pages (6,902 words), and the Discussion 6 pages (2,624 words).

The Results section reads more like a research report than a journal article.

It is recommended to reduce the length, particularly of the Results. This could be achieved by selecting key testimonies, since many can serve multiple analytical points. Another option is to omit results that have already been widely reported, such as “networking within a limited group,” “stigma and violence compel MSM to remain hidden,” and “condom-less sex”. These could simply be mentioned in the Discussion. Instead, the focus should be placed on what the paper truly contributes, such as new emerging MSM who remain hidden.

ANSWERING THE QUESTION: DO THE DATA AND ANALYSES FULLY SUPPORT THE CLAIMS? IF NOT, WHAT OTHER EVIDENCE IS REQUIRED?

Elements that should be reviewed and improved in the Methods

According to the Methods section of your article, the manuscript “[reference 13]” defines that “All study participants were adults and of legal age.”

Comment: While the mean age is provided, given the special relevance of participants’ age—especially concerning the topic of sugar daddies or uncles—it would be highly relevant to include a table with the sociodemographic characteristics of participants, including: age, type of informant (key informant or population), region of origin, rural/urban setting, and other characteristics that are pertinent for interpreting the findings.

The Methods section of your article also specifies: “MSM population categories recruited, and recruitment strategies are described in our previous report [reference 13].” However, in that study the dates and study sites differ (“four rural sites in Maharashtra, Odisha, Madhya Pradesh, and Uttar Pradesh between November 2018 and September 2019”).

Comment: It should be clarified whether this is the same study or a different one, given that this article includes more participants and more sites than the study mentioned in the Methods. If it is the same study, have the results of this article already been partially published before?

In the Methods section, it is stated: “Following purposive sampling techniques, 42 In-depth Interviews (IDIs), 40 Key Informant Interviews (KIIs), and 16 Focus Group Discussions (FGDs) leading to a total coverage among 98 participants, were conducted.”

Comment: It is not clear why both interviews and focus groups were conducted. Furthermore, of the 82 participants in interviews and the 16 FGDs, does the total of 98 participants include those who joined FGDs, or are they counted separately? It would be helpful to indicate how many individuals participated in FGDs and whether these were the same subjects who also participated in interviews.

In the Methods section, the article specifies: “Each participant was given a unique identification number which included alphabet/s abbreviations to identify state, Urban (U) or Rural (R) setting and type of data collection tool used and the number.”

Comment on identifiers: While U and R are clear, the other elements (e.g., MH in [##MH-U-IDI-03##]) should also be explained. It would be important to show how the identifiers are constructed and how they relate to key aspects of the analysis.

Moreover, considering the declared study objective (“to understand the population which is hidden, their issues and their expectations…”), the number of interviews (82) is strikingly high for a qualitative study, particularly with hard-to-reach populations. Therefore, it is important to clarify: whose voice are we hearing? Information such as the age of the informant, place of residence, and—if available—the behavioral profile that led to their inclusion, would greatly strengthen the analysis.

Additionally, conducting FGDs with highly stigmatized populations may itself contribute to stigma, as participants are required to share experiences in front of strangers. It would be important to explain: Why did you choose to conduct FGDs? How was confidentiality ensured for participants who, by definition, hide their identity for social and legal reasons?

Finally, was there any case in which a sugar daddy or uncle declared having sexual relations with minors? If so, how was this handled?

According to the Methods section, it is also stated: “Emerging themes were identified based on the categories formed following repeated iterations using the grounded theory approach [14].”

Comment: It would be valuable to explain how the major codes of analysis were configured, and whether any categories emerged during the study that had not been previously anticipated.

The Results section presents findings by broad themes but not by specific groups (hijra/transgender, Panthis, Kothis, or by age groups). If there are findings relevant to these categories, they should be highlighted.

Additionally, the first-time terms such as “chakka” or “kothi” appear, the meaning should be included in brackets.

It is also recommended to include COREQ (Consolidated Criteria for Reporting Qualitative Research) standards in the methodology.

COMMENTS ON CONTENT

On the treatment of child abuse in the context of previous literature on sexual violence and child abuse

Comment: The article clearly describes the presence of sugar daddies or adult men (50 years or older) who offer money to young people or children in exchange for unprotected or non-consensual sex. My central concern is with the way the data are named and analyzed, where consensual sex and coercive/forced sex are treated as equivalent “barriers” to HIV prevention, as if these practices were comparable in behavioral terms.

While all these behaviors may indeed result in similar prevention challenges, the lack of a critical position in the text and the “neutral” treatment of these practices risk inadvertently invisibilizing criminal behavior and sexual violence against minors.

International guidelines are clear: WHO, UNICEF, and UNAIDS recommend not using the terms “sexual debut” or “sexual initiation” when such events occur before the age of 18 through coercion, as this masks the violence. Instead, the recommended terms are sexual violence, child sexual abuse, or rape (when penetration and coercion occur). In HIV epidemiology, a distinction is explicitly made between consensual sexual debut and forced sexual debut, a category widely used in gender-based violence and sexual and reproductive health research.

The manuscript states: “In India, male child sexual abuse remains highly taboo, making acceptance and disclosure difficult.”

It appears that this taboo is reproduced, albeit unintentionally, in the manuscript, which does not explicitly denounce the violation of the human rights of society’s most vulnerable members—children. The text argues that adult men who sexually abuse children should be addressed through HIV prevention programs, while minors, victims of sexual assault, should be “empowered.” These statements should ideally be reviewed in light of human rights frameworks, child protection principles, and Indian law.

SPECIFIC EXAMPLES

• In the Abstract: “Both at urban and rural settings, the sexual debut of the child was between 2nd standard and 6th standard… The ‘sugar daddies/older MSM’ who were >50 years old seemed to slip from the continuum of programmatic prevention umbrella out of boredom, burn out, denial of self-risk, and the need to experiment with young boys.”

Comment: This phrasing should be reconsidered, as it risks “normalizing sexual violence against minors” as a “need.” Perhaps sugar daddies do not disengage from programs merely due to boredom or denial, but because they are abusing children.

• In the Discussion: “The study shows an emergence of critical themes which summarises the experiences of the hidden and complex vulnerabilities of boys and men.”

Comment: By referring to “complex vulnerabilities of boys and men”, the text appears to equate the vulnerabilities of minors with those of 50-year-old adults. While both groups may experience vulnerability, children are not only vulnerable—they are being victimized by adults. The absence of explicit recognition could risk invisibilizing this, thereby exacerbating the vulnerability of abused children.

• In the Discussion: “As reported in several studies from India and elsewhere, our study confirms a very young age of sex initiation (6–7 years), obviously condom-less. This early sexual debut (consensual/coercive/exploited) has been associated with increased risk of unintended pregnancy, STIs and future risky sexual behaviour.”

Comment: Placing “consensual, coercive, or exploited” in parentheses, separated by commas, makes them appear as variants of the same phenomenon, when in reality they are fundamentally different. Referring to rape or human sexual exploitation as “sexual debut” is problematic and should be revised.

• In the Discussion: “The early initiation of adolescent MSM and their sexual interactions with sugar daddies underscores the urgent need to empower and educate young boys…”

Comment: Here again, the term “empower” is not appropriate. Children and adolescents must be protected by law, not “empowered or educated.” Empowerment implies restoring agency to a subject capable of exercising it, but a minor cannot exercise power in such circumstances. Therefore, the appropriate response is protection, not empowerment.

• Conclusions and Recommendations: The manuscript states that MSM-focused services with affirmative care, peer referral models, and mental health counseling could facilitate access among hidden MSM, and that “there is an urgent need to revisit strategies for engaging middle-aged MSM, the ‘sugar daddies’ or ‘uncles.’”

Comment: This treats sugar daddies merely as individuals in need of prevention or mental health programs, while overlooking the criminal nature of engaging in sexual relations with minors.

ON THE INTRODUCTION

The Introduction states: “The Indian National Strategic Plan for HIV/AIDS and STI (2017-24) focuses on Community Strengthening Strategy (CSS) with definite roles for KPs who are the centre of Community Strengthening activities of the HIV prevention and control program [11].”

Comment: It would be very valuable if the authors could critically articulate the Plan’s position on child sexual abuse and exploitation, and provide recommendations in this regard.

FINAL RECOMMENDATION

If the paper is considered unsuitable for publication in its present form, the study nevertheless shows sufficient potential to merit revision and resubmission. It is recommended that the methodological aspects noted above be clarified, and that the manuscript be reframed through the lens of human rights, sexual violence, and child protection. This includes making visible the issue of sexual abuse and clearly distinguishing consensual sex from coerced or exploitative sex throughout the analysis.

**Do you want your identity to be public for this peer review?** For information about this choice, including consent withdrawal, please see our Privacy Policy

Reviewer #1: **Yes:** Ricardo Baruch

Reviewer #2: No

---

## [Author Response · Author response to Decision Letter 1]

10 Nov 2025

We thank the Editor and reviewers for their valuable feedback. We sincerely appreciate the reviewers’ critical insights, which have helped us re-examine our analysis and discussion in depth. We have revised the manuscript to place a stronger focus on sexual violence within the context of child abuse. The neutral treatment of these practices has been removed.

We have also clearly distinguished between consensual and forced sexual debut, and revised the manuscript in light of human rights frameworks, child protection principles, and relevant Indian law.

Once again, we thank the Editor and reviewers for their thoughtful comments and guidance.

A detailed, point-by-point response to all reviewer and editor comments is provided in the separate document titled “Response to Reviewers.”

All corresponding revisions have been incorporated in the “Revised Manuscript with Track Changes.”

---

## [Decision Letter · Decision Letter 1]

18 Dec 2025

The hidden/ hard to reach Men who have Sex with Men (MSM) – Results from the qualitative study in India

PONE-D-25-09473R1

Dear Dr. Sahay,

We’re pleased to inform you that your manuscript has been judged scientifically suitable for publication and will be formally accepted for publication once it meets all outstanding technical requirements.

Kind regards,

Juan Pablo Gutierrez

Academic Editor

PLOS One

Additional Editor Comments (optional):

Reviewers' comments:

Reviewer's Responses to Questions

**Comments to the Author**

Reviewer #2: All comments have been addressed

2. Is the manuscript technically sound, and do the data support the conclusions?

Reviewer #2: Yes

3. Has the statistical analysis been performed appropriately and rigorously?

Reviewer #2: N/A

4. Have the authors made all data underlying the findings in their manuscript fully available?

Reviewer #2: No

5. Is the manuscript presented in an intelligible fashion and written in standard English?

Reviewer #2: Yes

Reviewer #2: I wish to begin by acknowledging the substantial and constructive work undertaken in revising this manuscript. The reorganization, clarification of key concepts, and methodological refinements have significantly enhanced the overall quality of the paper. I commend the authors for the clarity, coherence, and analytic depth achieved in this revised version, as well as for the originality of their contributions—particularly the emphasis on the extreme vulnerability of very young adolescents and the urgent need to broaden prevention frameworks to adequately address this population.

I have carefully reviewed and validated the modifications made throughout the manuscript. The improvement of the study objective, the inclusion of detailed sociodemographic tables, and the revision of quotation identifiers (now including type of informant and age) are all important advances that strengthen both transparency and interpretability. The clarification provided in the Methods and Discussion sections regarding previously published components of the project is appropriate and necessary. I would only recommend that, if any specific results have been disseminated elsewhere, these be removed from the current manuscript to maintain full alignment with PLOS ONE’s originality criteria.

From an analytical standpoint, the results section now more clearly delineates differences across MSM subgroups, enhancing the reader’s ability to understand their distinct contexts, vulnerabilities, and programmatic implications. I particularly commend the authors for eliminating the previously neutral tone regarding child sexual abuse and for making a clear conceptual distinction between consensual sexual behavior and coercive or violent acts. This revision is not only ethically essential but also elevates the scientific relevance of the study by accurately framing one of its central findings. It is my view that the manuscript now articulates these sensitive issues with appropriate rigor, specificity, and public health relevance.

The manuscript now meets the journal’s expectations regarding methodological clarity, analytic robustness, ethical transparency, and contribution to the field. The study offers meaningful insights into hidden and hard-to-reach MSM populations and provides valuable evidence with direct implications for HIV prevention policy and programming.

In its current form, I consider the manuscript suitable for publication, subject only to the minor recommendation noted above. I congratulate the authors for the substantial improvements made and for producing a manuscript that will undoubtedly contribute to advancing knowledge and informing practice in this critical area.

**Do you want your identity to be public for this peer review?** For information about this choice, including consent withdrawal, please see our Privacy Policy

Reviewer #2: No

---

## [Editor Report · Acceptance letter]

PONE-D-25-09473R1

PLOS One

Dear Dr. Sahay,

I'm pleased to inform you that your manuscript has been deemed suitable for publication in PLOS One. Congratulations! Your manuscript is now being handed over to our production team.

Kind regards,

on behalf of

Dr. Juan Pablo Gutierrez

Academic Editor

PLOS On